# Molecular Signatures of Dendritic Cell Activation upon TNF Stimulation: A Multi-Omics Study in Rheumatoid Arthritis

**DOI:** 10.3390/ijms26136071

**Published:** 2025-06-24

**Authors:** Alina Alshevskaya, Shakir Suleimanov, Elizaveta Sheveleva, Roman Perik-Zavodskii, Olga Perik-Zavodskaia, Saleh Alrhmoun, Julia Lopatnikova, Julia Zhukova, Nadezhda Shkaruba, Natalia Sivitskaya, Alexey Sizikov, Elena Golikova, Sergey Sennikov

**Affiliations:** 1Laboratory of Immune Engineering, Federal State Autonomous Educational Institution of Higher Education, I.M. Sechenov First Moscow State Medical University of the Ministry of Health of the Russian Federation (Sechenov University), 119048 Moscow, Russia; suleymanovef@gmail.com (S.S.); saleh.alrhmoun1@gmail.com (S.A.); tamah@inbox.ru (E.G.); 2Laboratory of Molecular Immunology, Federal State Budgetary Scientific Institution Research Institute of Fundamental and Clinical Immunology (RIFCI), 630099 Novosibirsk, Russia; perik.zavodskaia@gmail.com (O.P.-Z.);; 3Rheumatology Department of the Immunopathology Clinic, Federal State Budgetary Scientific Institution Research Institute of Fundamental and Clinical Immunology, 630099 Novosibirsk, Russia

**Keywords:** single-cell, multi-omics, CITE-seq, rheumatoid arthritis, dendritic cells, TNF-alpha, TNFR1, TNFR2, inflammation, immune regulation

## Abstract

Dendritic cells (DCs) play a central role in the immunopathogenesis of rheumatoid arthritis (RA), yet their regulation by tumor necrosis factor alpha (TNF) and associated receptors remains poorly characterized. We applied a single-cell multi-omics approach (CITE-seq) to profile peripheral blood mononuclear cells (PBMCs) from RA patients and healthy donors, before and after in vitro TNF stimulation. Using integrated analysis of surface protein expression and transcriptomic data, we focused on phenotypic and transcriptional changes in dendritic cell populations. DCs from RA patients exhibited elevated surface expression of CD14 and CD16, indicative of an inflammatory phenotype, and showed marked responsiveness to TNF. Upon stimulation, RA-derived DCs upregulated genes involved in antigen presentation (*CD83*, *LAMP3*), lymph node migration (*CCR7*, *ADAM19*), and inflammation (*TRAF1*, *IL24*) whereas such activation was absent in healthy controls. Our data reveal a TNF-responsive, pro-inflammatory transcriptional program in dendritic cells from RA patients and underscore the relevance of the TNF receptor profile in shaping DC function. These findings provide new insights into the immunobiology of RA and identify dendritic cells as potential targets for personalized immunomodulatory therapy.

## 1. Introduction

Dendritic cells (DCs) serve as central regulators of immune responses, functioning as professional antigen-presenting cells that orchestrate the activation and polarization of T lymphocytes. Their involvement in the pathogenesis of rheumatoid arthritis (RA) is well established and is mediated through the activation of autoreactive T cells, the production of pro-inflammatory cytokines, and the coordination of local immune cell interactions, collectively sustaining a chronic inflammatory microenvironment [1,2,3]. TNF-α plays a central role not only in RA but also in related immune-mediated disorders, including psoriasis and psoriatic arthritis (PsA), where shared inflammatory pathways drive both skin and joint pathology [4]. In RA, anti-TNF agents such as infliximab and adalimumab effectively suppress synovial inflammation and prevent joint destruction [5], while in PsA and psoriasis they also reduce cutaneous lesions and halt structural progression [6]. The responsiveness of dendritic cells to TNF signaling may contribute to therapeutic efficacy across these conditions, particularly by modulating DC–T cell interactions and the inflammatory milieu in both joint and skin environments.

In the synovial tissue of RA patients, various subsets of DCs—including monocyte-derived DCs (moDCs), plasmacytoid DCs (pDCs), and conventional CD1c^+^ DCs—are present at increased frequencies. These cells contribute to both the initiation and maintenance of inflammation by secreting cytokines such as interleukin-6 (IL-6), tumor necrosis factor alpha (TNF), and IL-1β [2,7]. Furthermore, DCs play a pivotal role in promoting Th17 cell differentiation and disrupting immune tolerance, both of which are central to the development of autoimmune pathology in RA [8,9]. Certain DC subsets, particularly plasmacytoid DCs, have demonstrated the capacity to acquire tolerogenic properties, highlighting their potential as targets for antigen-specific immunotherapeutic approaches [10,11].

The functional properties of DCs are regulated by a range of cytokines, with TNF representing a key modulatory factor. TNF exerts its effects through two distinct receptors: TNFR1 and TNFR2 [12]. TNFR1 is broadly expressed across multiple cell types and primarily activates pro-inflammatory signaling pathways and apoptosis, whereas TNFR2, which is predominantly restricted to immune cells, is associated with the expansion of regulatory T cells, the promotion of immune homeostasis, and cell survival [13,14,15,16].

Despite extensive research into TNF signaling pathways [17], the molecular mechanisms that govern dendritic cell responses to TNF stimulation in RA remain incompletely understood. In particular, the impact of TNFR1 and TNFR2 expression and co-expression on the transcriptional and functional profiles of DCs under inflammatory conditions has not been fully elucidated. Recent studies have demonstrated that the co-expression profiles of TNFR1 and TNFR2 on immunocompetent cells are significantly altered in RA and are associated with disease activity and clinical severity indicators [18]. These observations suggest that the balance between TNFR1 and TNFR2 expression critically influences immune cell function in RA; however, the specific contribution of these receptors to dendritic cell activation remains to be clarified.

Recent advances in single-cell multi-omics technologies, such as cellular indexing of transcriptomes and epitopes by sequencing (CITE-seq), now enable the simultaneous profiling of gene expression and surface protein markers at single-cell resolution, providing unprecedented opportunities to dissect the molecular states of immune cells in complex disease settings [19,20].

In the present study, we employed a single-cell multi-omics approach to investigate the phenotypic and transcriptomic responses of peripheral blood dendritic cells from healthy individuals and patients with RA following TNF stimulation, with a particular focus on how TNF receptor expression profiles modulate dendritic cell activation.

## 2. Results

### 2.1. Identification and Annotation of the Dendritic Cell Population

As part of the single-cell multi-omics analysis, peripheral blood mononuclear cells (PBMCs) from three healthy donors and three patients with RA were analyzed before and after stimulation with TNF. Cell types were annotated based on surface protein expression and canonical marker genes.

Particular focus was placed on the DC population, which was clearly distinguished in the UMAP embedding derived from weighted nearest neighbor (WNN) integration of transcriptomic and surface protein data (Figure 1A). The distribution of cells within the DC cluster was balanced across donor groups, indicating successful batch effect correction (Figure 1B).

### 2.2. Expression of TNF Receptors on the Surface of Dendritic Cells

Visualization of surface protein levels of TNF receptors (TNFR1p and TNFR2p) revealed a general downregulation of both markers following TNF stimulation in both healthy donors and RA patients (Figure 1C, upper panel). However, the magnitude of downregulation was notably lower in RA patients compared to healthy controls.

In addition, DCs from RA patients exhibited higher surface expression of CD14p and CD16p proteins, which may indicate a shift toward a pro-inflammatory phenotype (Figure 1C, upper panel).

No significant differences were observed at the transcriptomic level in the expression of TNFRSF1A and TNFRSF1B—the genes encoding TNFR1 and TNFR2, respectively—between groups (Figure 1C, lower panel), suggesting that the observed changes in surface receptor expression likely occur through post-transcriptional regulatory mechanisms. A similar pattern was observed for CD14 and FCGR3A gene expression (Figure 1C, lower panel).

### 2.3. TNFR1/TNFR2 Ratio and Receptor Profile Characteristics

Further analysis of the relative surface expression of TNFR1p and TNFR2p revealed that TNFR2p predominated in most immune cell subsets, including dendritic cells (Figure 2). However, within the dendritic cell population, surface levels of TNFR1p and TNFR2p were largely comparable in both groups, with a slight shift toward TNFR2p in RA patients (Figure 1D). This ratio may reflect altered cellular sensitivity to regulatory or pro-inflammatory signaling in the context of chronic inflammation in RA.

### 2.4. Differential Gene Expression in Dendritic Cells Before and After TNF Stimulation

To identify transcriptional programs induced by TNF stimulation, differential gene expression analysis was performed using a pseudobulk approach separately for each study group. In healthy donors, stimulation induced minimal changes, except for increased expression of MIR155HG, a gene associated with microRNA regulation and potential immunoactivation.

In contrast, RA patients exhibited significant upregulation of multiple genes associated with dendritic cell activation, migration, and pro-inflammatory function following TNF stimulation (Figure 3):CD83 and LAMP3: markers of dendritic cell maturation involved in antigen presentation [21,22];CCR7 and ADAM19: genes mediating lymph node homing and migratory capacity [23];TRAF1 and IL24: components of inflammatory signaling cascades linked to chronic inflammation [24,25,26];ANXA1: a protein modulating dendritic cell–T cell interactions, potentially contributing to T cell proliferation [27].

**Figure 3 ijms-26-06071-f003:**
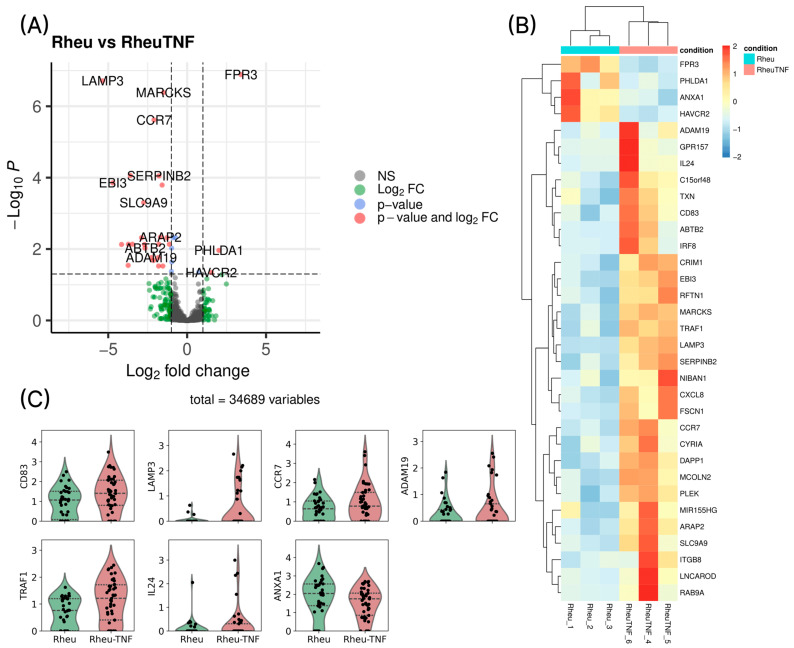
Differential gene expression analysis. (**A**) Volcano plot showing differential gene expression between TNF-stimulated and unstimulated dendritic cells. Green dots represent genes with |log_2_ fold change| < 0.58; blue dots correspond to genes with *p*-value > 0.05; red dots indicate significantly up- or downregulated genes with |log_2_ fold change| > 0.58 and adjusted *p*-value < 0.05; grey dots represent non-significant changes. (**B**) Heatmap displaying the expression levels of differentially expressed genes across experimental groups. Red indicates higher expression; blue indicates lower expression. (**C**) Violin plots showing the expression dynamics of selected genes in dendritic cells from RA patients (Rheu), before and after TNF stimulation. Norm—healthy donors; Rheu—rheumatoid arthritis patients; TNF—tumor necrosis factor alpha.

## 3. Discussion

The findings of this study demonstrate that dendritic cells (DCs) from patients with rheumatoid arthritis (RA) exhibit not only an altered surface phenotype but also heightened responsiveness to tumor necrosis factor alpha (TNF) stimulation. This is reflected in both the expression of surface markers and the activation of specific transcriptional programs.

One of the key observations was the increased expression of CD14 and CD16 on the surface of DCs from RA patients. These markers are typically associated with inflammatory myeloid cells and are linked to the production of pro-inflammatory cytokines and enhanced interactions with T cells [1,3]. This pre-activated phenotype is further supported by the elevated expression of the maturation markers CD83 and LAMP3 following TNF stimulation. In addition to its role as a co-stimulatory molecule, CD83 also functions as a biomarker with the capacity to modulate both inflammatory and tolerogenic responses [22]. LAMP3^+^ DCs, as shown in recent cancer immunology studies, contribute to immune regulation by promoting CD8^+^ T cell exhaustion and recruiting regulatory T cells (Tregs), thereby establishing an immunosuppressive microenvironment [21]. While these findings were obtained in the context of cancer, they point to a potential immunomodulatory role for LAMP3^+^ DCs that may be relevant in autoimmune inflammation such as RA.

The upregulation of CCR7 following TNF stimulation suggests an enhanced migratory potential of DCs. CCR7 directs the trafficking of antigen-presenting cells to lymph nodes, where naive T cells are primed [23]. Notably, CCR7 plays dual roles in promoting both immune activation and tolerance depending on the cellular and microenvironmental context. The observed shift toward CCR7 expression may indicate an attempt by DCs to enter a lymphoid presentation mode, which, under chronic inflammatory conditions such as RA, may facilitate sustained immune activation.

Of particular interest is the upregulation of the inflammatory mediators TRAF1 and IL24. TRAF1 is an adaptor protein in TNF receptor-associated signaling pathways and also modulates the Toll-like receptor (TLR) and NF-κB activation cascades [25]. Genetic variations in TRAF1 have been linked to an increased risk of RA, and experimental disruption of TRAF1/cIAP2 interactions has been shown to attenuate inflammation and disease severity in arthritis models [26]. The function of TRAF1 may be protective or pathological depending on the balance of TNFR1/TNFR2-mediated signals. In this study, its upregulation following TNF stimulation supports the enhancement of a pro-inflammatory program in DCs. The relevance of TNF signaling extends beyond RA, as shared transcriptional and inflammatory pathways are also implicated in PsA and Pso, where anti-TNF therapies show comparable efficacy in modulating DC function and disease outcomes.

Another important observation was the upregulation of IL24, a cytokine belonging to the IL-10 family, in RA patients following TNF stimulation. IL24 is known to exhibit a dual role: it can contribute to both the induction of inflammation and the regulation of immune responses [24]. In the context of RA, increased IL24 expression has been associated with the activation of innate immunity and tissue damage. However, depending on the surrounding receptor landscape, IL24 may also promote tolerogenic responses, suggesting its potential as a regulatory mediator under specific conditions.

Another important finding concerns the expression of ANXA1 (Annexin A1), a protein involved in the regulation of dendritic cell–T cell interactions. In our study, ANXA1 expression was decreased in dendritic cells from RA patients following TNF stimulation. Previous work by Huggins et al. demonstrated that the absence of ANXA1 in dendritic cells leads to impaired upregulation of maturation markers, reduced migratory activity, attenuated production of pro-inflammatory cytokines (IL-1β, TNF-α, and IL-12), and diminished activation of key signaling pathways such as NF-κB, ERK1/2, and Akt [27]. These functional defects ultimately result in a reduced ability of dendritic cells to stimulate T-cell proliferation. Our data are consistent with these observations and suggest that downregulation of ANXA1 in RA may contribute to altered maturation and functional impairment of dendritic cells in the inflammatory environment.

The transcriptional changes identified in TNF-stimulated dendritic cells from RA patients suggest enhanced potential to modulate the DC–T cell axis through the coordinated upregulation of key immune-regulatory molecules. Specifically, CD83, upregulated upon TNF stimulation, is a maturation marker whose expression promotes effective T cell activation and is tightly regulated at the post-transcriptional level [28]. Increased CCR7 expression suggests enhanced DC migration to lymph nodes and coordination of chemokine receptor reprogramming in interacting T cells [29]. Additionally, IL24, induced in RA DCs, may modulate T cell responses by influencing cytokine-mediated feedback loops, potentially linking TNF-induced activation to broader immune polarization [30]. Together, these shifts indicate that TNF-triggered transcriptional programs in RA DCs—potentially shaped by altered TNFR1/TNFR2 expression—enhance their ability to orchestrate adaptive immune responses and sustain inflammation.

Finally, it should be noted that alterations in the surface receptor profile of dendritic cells (DCs) may reflect disruptions in immune homeostasis. Specifically, DCs from patients with rheumatoid arthritis (RA) exhibit an increased expression of CD14 and CD16 proteins on their surface, indicating a shift towards a pro-inflammatory phenotype. CD14+ and CD16+ cells are characterized by the increased expression of pro-inflammatory markers and the production of pro-inflammatory cytokines, as well as the ability to stimulate Th17 responses, all of which contribute to the pathogenesis of autoimmune inflammation in RA [31].

This study has some limitations. The number of enrolled donors was small, and only peripheral blood mononuclear cells were analyzed, which may not fully reflect the tissue-specific immune environment of rheumatoid arthritis. Additionally, in vitro TNF stimulation does not entirely replicate the complex cellular interactions and mechanical context of the synovial niche. Nonetheless, our single-cell multi-omics approach enabled high-resolution profiling of over 30,000 immune cells, including hundreds to thousands of dendritic cells per individual, which partially offsets the limited cohort size. While our study was not designed to assess clinical outcomes, the observed TNF-induced transcriptional programs and receptor expression patterns provide candidate molecular signatures that may inform the future development of personalized immunomodulatory strategies.

Our single-cell multi-omics analysis provided detailed phenotypic and transcriptional characterization of dendritic cells in RA, uncovering specific alterations associated with their activation in response to TNF stimulation. Despite the robustness of the findings, further studies are needed to validate these results in independent cohorts and to investigate local immune responses in the synovial tissue. Future directions should also include functional analyses of the identified molecular targets and their roles in intercellular communication, particularly in the context of T cell-mediated immunity. Taken together, these data lay a foundation for the further exploration of dendritic cells in RA pathogenesis and may inform the development of novel immunotherapeutic strategies.

## 4. Materials and Methods

### 4.1. Cell Preparation

Peripheral blood samples were collected from healthy donors and patients with confirmed rheumatoid arthritis (RA) treated at the Clinic of Immunopathology, Research Institute of Fundamental and Clinical Immunology (Novosibirsk, Russia). The study cohort included three female RA patients aged 57–67 years, all seropositive for rheumatoid factor and anti-cyclic citrullinated peptide antibodies and presenting with high disease activity (DAS28 scores ranging from 5.17 to 6.69). All patients were undergoing treatment with conventional disease-modifying antirheumatic drugs (methotrexate or leflunomide). The control group consisted of three age-matched, conditionally healthy women (aged 62–68 years) without any rheumatologic conditions. All donors and patients provided their written informed consent to participate in the study. The research was performed in accordance with the Declaration of Helsinki and was approved by the local ethics committee of the FSBI ‘Research Institute of Clinical Immunology’ (protocol no. 145, dated 19 April 2024).

Up to 9 mL of peripheral blood was collected from the cubital vein under sterile conditions into vacuum tubes containing K_3_-EDTA (Greiner Bio-One GmbH, Kremsmünster, Austria). Peripheral blood mononuclear cells (PBMCs) were isolated using Ficoll–Urografin density gradient centrifugation (1.077 g/mL; PanEko, Moscow, Russia). After two washes in PBS, the cells were cryopreserved in a freezing medium containing 90% fetal bovine serum (FBS) and 10% DMSO.

On the day of the experiment, the cryopreserved cells were thawed and transferred into RPMI-1640 medium (BioloT, Voronezh, Russia) supplemented with 30% FBS (HyClone, Logan, UT, USA). After two rounds of centrifugation at 1500 rpm for 10 min, the cells were resuspended at a concentration of 1 × 10^6^ cells/mL and incubated for 6 h with recombinant TNF (5 ng/μL) in RPMI-1640 medium supplemented with 10% FBS, 2 mM L-glutamine (BioloT, Voronezh, Russia), 5 × 10^−4^ M 2-mercaptoethanol (Sigma-Aldrich, St. Louis, MO, USA), 80 µg/mL gentamicin (KRKA, Novo Mesto, Slovenia), 10 mM HEPES (Sigma-Aldrich), and 100 µg/mL benzylpenicillin (Biosintez, Penza, Russia).

### 4.2. Sample Tag and AbSeq Labeling

Following TNF stimulation, peripheral blood mononuclear cells (PBMCs) were stained with BD™ Single-Cell Multiplexing Sample Tag antibodies (BD Biosciences, San Jose, CA, USA, Cat# 633781) to enable sample barcoding, and with a 10-plex AbSeq antibody panel targeting key immune surface markers (CD4, CD8, CD14, CD16, CD19, CD45RA, CD45RO, CD56, TNFR1, and TNFR2; BD Biosciences, San Jose, CA, USA) to enable simultaneous protein expression profiling. Staining was performed for 30 min at room temperature according to the manufacturer’s instructions. The specificity of AbSeq antibodies for TNFR1 and TNFR2 was qualitatively validated prior to analysis by comparing their binding profiles to those of conventional fluorochrome-conjugated monoclonal antibodies targeting TNFRSF1A and TNFRSF1B. Across replicate tests, the AbSeq reagents showed consistent signal localization and, on average, 7–9% higher sensitivity than conventional antibodies when detecting low-abundance surface expression. These findings confirmed the suitability of the AbSeq panel for high-resolution detection of TNF receptor proteins in our system. After three washing steps, cells were labeled with Calcein and counted using the Attune NxT flow cytometer. Samples were then pooled in equal proportions and resuspended at 30 cells/μL in cold sample buffer. The final cell suspension was loaded onto two BD Rhapsody cartridges for single-cell capture. PBMCs from healthy donors and RA patients were processed in parallel to minimize batch effects.

### 4.3. Library Preparation and Sequencing

Single-cell RNA and protein libraries were prepared using the BD Rhapsody Express Single-Cell Analysis System following the manufacturer’s protocols for Whole Transcriptome Analysis (WTA), AbSeq, and Sample Tag libraries. Briefly, mRNA and oligo-conjugated antibodies were captured on magnetic beads, reverse transcribed, and amplified through sequential PCR steps with quality control at each stage. Final libraries were quantified with a Qubit 4 Fluorometer and quality-checked via capillary electrophoresis (Qsep1). Libraries were pooled at an approximate 82:16:2 ratio for WTA, AbSeq, and Sample Tag libraries, respectively, and sequenced on the NovaSeq 6000 system (Illumina) using an S1 flow cell (paired-end reads, 1300 million total reads, R1 = 71, R2 = 51 cycles).

### 4.4. Data Preprocessing

We processed the FASTQ files obtained from sequencing using the BD Rhapsody pipeline v2.0 (BD Biosciences). The general workflow for data preprocessing and integration was adapted from best practices established in recent large-scale single-cell immuno-oncology studies [32]. The pipeline filtered out low-quality reads based on length, base quality, and nucleotide frequency, then identified cell barcodes and UMIs in high-quality R1 reads. R2 reads were aligned to transcriptome (STAR) and AbSeq (Bowtie2) references. Reads with matching barcodes, UMIs, and genes were collapsed, and counts were corrected for errors (RSEC for both, DBEC for AbSeq). Cell counts were estimated by second-derivative analysis to filter noise. Multiplexed samples were demultiplexed and multiplets removed using Sample Tag antibodies. The pipeline identified 31,268 single cells (2500–3500 per sample) and output gene and surface protein expression matrices. Sequencing metrics showed 92–98% saturation and an average RSEC depth of 5.5 (above medium).

Data preprocessing was performed using the Scanpy package (version 1.9.1) [33]. Initially, cells with fewer than 500 or more than 7000 detected genes, as well as those containing over 25% mitochondrial RNA content, were excluded. Genes expressed in fewer than three cells were also removed. Potential RNA-based doublets were identified and excluded using Scrublet [34]. Additionally, cells with total surface protein expression values exceeding 7500 were filtered out.

RNA expression data were normalized for library size and log-transformed (log-normalization) using normalize_total followed by a log1p transformation. Additional Pearson residual normalization with normalize_pearson_residuals was performed for the identification of highly variable genes (HVGs). Protein expression data were normalized using centered log-ratio (CLR) transformation, using the clr function.

Dimensionality reduction was carried out independently for both RNA and protein modalities using principal component analysis (PCA). For RNA modality, top 3000 HVGs were selected using the highly_variable_genes. Principal component analysis (PCA) was then performed on the scaled data independently for both RNA and protein modalities. Batch effects were corrected using the Harmony algorithm (harmony_integrate Scanpy function) independently for both modalities [35].

K-nearest neighbors (KNN) graphs were constructed separately for each modality using the neighbors function, and integration of RNA and protein modalities was performed using the Weighted Nearest Neighbors (WNN) algorithm with the neighbors function in the Muon package [36]. Based on the resulting WNN graph, clustering was conducted using the Leiden algorithm, and dimensionality reduction for visualization was applied via Uniform Manifold Approximation and Projection (UMAP) implemented in the Scanpy function umap. Cell clusters were manually annotated based on the expression of canonical marker genes and surface proteins.

### 4.5. Differential Gene Expression Analysis

Differential gene expression analysis was performed between unstimulated and TNF-stimulated conditions using a pseudobulk aggregation approach implemented with the Python tool adpbulk (version 0.1.4). Raw counts were aggregated across all cells within each defined cell population. Differential expression analysis was performed using the DESeq2 package [37]. Volcano plots were generated using the EnhancedVolcano R package (version 1.24.0). Genes were considered differentially expressed if they exhibited a log_2_ fold change greater than 1 and an adjusted *p*-value below 0.05.

## 5. Conclusions

Dendritic cells from patients with RA exhibit a pronounced pro-inflammatory phenotype and heightened sensitivity to TNF stimulation. Using single-cell multi-omics analysis, we identified the activation of transcriptional programs in RA patients following TNF stimulation that are associated with antigen presentation (CD83, LAMP3), migration to lymphoid tissues (CCR7, ADAM19), and the production of pro-inflammatory mediators (TRAF1, IL24, ANXA1). The observed upregulation of surface CD14p and CD16p in RA patients may reflect an impaired immunoregulatory capacity. Altogether, our findings highlight the functional importance of dendritic cells as active contributors to the inflammatory cascade in RA and suggest their potential as a source of biomarkers and targets for personalized therapeutic strategies. These molecules may serve as biomarkers of dendritic cell activation in RA and warrant further validation in larger patient cohorts. Future studies with larger and longitudinally followed cohorts are essential to confirm the clinical relevance of our findings.

## Figures and Tables

**Figure 1 ijms-26-06071-f001:**
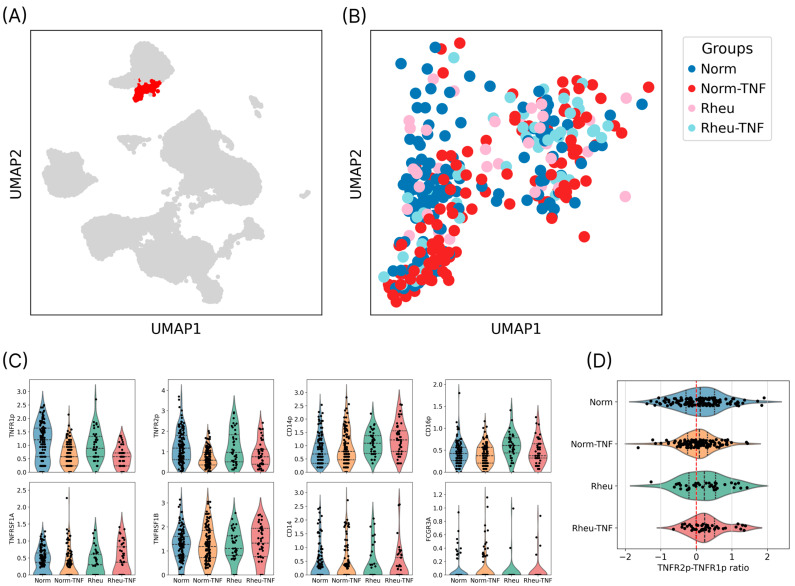
Identification of the dendritic cell cluster and analysis of TNF receptor expression. (**A**) UMAP projection of all cells, highlighting the dendritic cell cluster in red. (**B**) UMAP projection showing the distribution of dendritic cells across experimental groups. (**C**) Violin plots showing surface protein expression levels of TNFR1p and TNFR2p (**top panel**) and transcript levels of TNFRSF1A and TNFRSF1B (**bottom panel**) in dendritic cells from each group. (**D**) Violin plot of TNFR2/TNFR1 surface protein expression ratio in dendritic cells from RA patients (Rheu) and healthy donors (Norm), before and after TNF stimulation (TNF).

**Figure 2 ijms-26-06071-f002:**
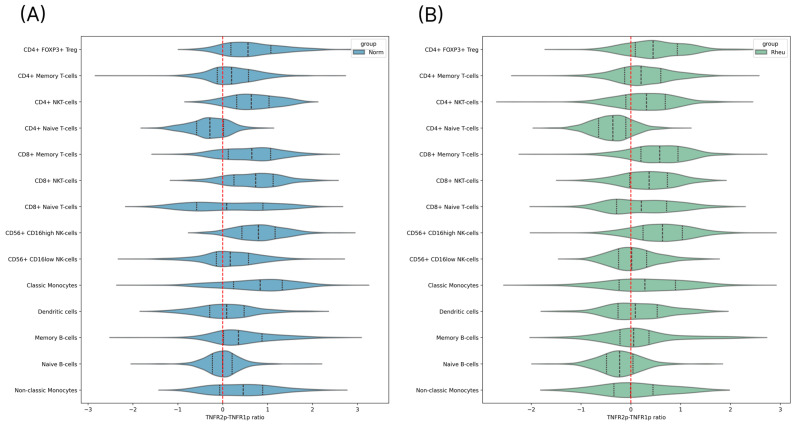
Analysis of relative TNFR1p and TNFR2p expression. Violin plots showing the TNFR2/TNFR1 surface expression ratio across immune cell types in the healthy donor group (**A**) and in the rheumatoid arthritis patient group (**B**).

## Data Availability

The datasets presented in this study are publicly available in the NCBI Gene Expression Omnibus (GEO) repository under accession number GSE289019 (https://www.ncbi.nlm.nih.gov/geo/query/acc.cgi?acc=GSE289019 (accessed on 16 June 2025)).

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
