# Peer review of "Molecular Signatures of Dendritic Cell Activation upon TNF Stimulation: A Multi-Omics Study in Rheumatoid Arthritis"

_ijms, 2025, doi:10.3390/ijms26136071_

Round 1

Reviewer 1 Report

Comments and Suggestions for Authors

Alshevskaya explored the role of TNF-regulated dendritic cells (DCs) in the immunopathogenesis of rheumatoid arthritis (RA) by single-cell transcriptome and proteome. The design of the cohort study is sound, yet the process of data processing still requires specific and detailed disclosures that enhance the reproducibility of the study.

  1. Discussion. Specific mechanisms of regulation and potential cellular communication links need to be added further.
  2. The data processing procedures of M&M. single-cell transcriptome and metabolome are still insufficient, please refer to the related articles of Prof. Zhang Zemin's group in Peking University for details.
  3. If possible, please submit raw data in a public database. The public availability of relevant data benefits the advancement of science worldwide.

Author Response

Comment 1: Discussion. Specific mechanisms of regulation and potential cellular communication links need to be added further.

Answer: Thank you for pointing out the need to expand on regulatory mechanisms and cell-cell communication. We have added relevant interpretations to the Discussion section, particularly emphasizing how TNF-induced transcriptional changes in DCs may affect T cell priming and immune polarization (highlighted in yellow).

Comment 2: The data processing procedures of M&M. single-cell transcriptome and metabolome are still insufficient, please refer to the related articles of Prof. Zhang Zemin's group in Peking University for details.

Answer:  We appreciate the suggestion and have expanded the Materials and Methods section 4.4 to provide a more detailed description of our single-cell data processing, including references to commonly accepted pipelines similar to Zhang Zemin's group (highlighted in yellow).

Comment 3: If possible, please submit raw data in a public database. The public availability of relevant data benefits the advancement of science worldwide.

Answer: Thank you for raising this important point. At the time of manuscript submission, the dataset was undergoing deposition. It is now publicly available in the NCBI GEO repository under accession number GSE289019. The Data Availability Statement has been updated accordingly.

Reviewer 2 Report

Comments and Suggestions for Authors

comments:

nice abstract

report more on the role of anti-TNF in case of RA as well as tHE role of TNF in co-existing Pso and PsA

Provide limitations and clinical implications ( maybe a personalised approach of such patients to use specific drugs)

well presented figures and methods

generally a well-written manuscript 

Author Response

Comment 1: report more on the role of anti-TNF in case of RA as well as tHE role of TNF in co-existing Pso and PsA

Answer: Thank you for pointing this out. We have expanded the Introduction to briefly reflect the broader relevance of TNF signaling and anti-TNF therapy across RA and related conditions such as PsA and psoriasis.

Comment 2: Provide limitations and clinical implications ( maybe a personalised approach of such patients to use specific drugs)

Answer: Thank you for your comment. We have added a limitations paragraph to clarify that, while no clinical interventions were based on this analysis, the dataset provides a high-resolution view of dendritic cell activation and suggests directions for future biomarker-guided strategies.

Comments 3 and 4: well presented figures and methods. Generally a well-written manuscript 

Answer: We sincerely thank the reviewer for the positive feedback on the quality of the figures, methods, and overall presentation of the manuscript.

Reviewer 3 Report

Comments and Suggestions for Authors

This study employs a single-cell multi-omics approach (CITE-seq) to investigate the phenotypic and transcriptional responses of dendritic cells (DCs) in patients with rheumatoid arthritis (RA) following tumor necrosis factor-alpha (TNF) stimulation. 

The study includes only three RA patients and three controls. While the single-cell approach yields large datasets, the extremely limited number of biological replicates limits generalizability and robustness.

The use of in vitro TNF stimulation may not fully recapitulate the complex in vivo inflammatory milieu of RA joints.

No experimental validation of observed transcriptional changes (e.g., functional assays or protein expression validation beyond AbSeq) was conducted.

The study introduces several candidate biomarkers (CD83, LAMP3, IL24) and immunoregulatory shifts, providing leads for future investigation.

However, the limited cohort size tempers the impact. Larger, longitudinal studies with functional follow-up are needed to establish clinical utility.

Author Response

Comment 1: The study includes only three RA patients and three controls. While the single-cell approach yields large datasets, the extremely limited number of biological replicates limits generalizability and robustness.

Answer: Thank you for pointing this out. We fully acknowledge this limitation and have addressed it in the revised Discussion section.

Comment 2: The use of in vitro TNF stimulation may not fully recapitulate the complex in vivo inflammatory milieu of RA joints.

Answer: We agree, and this is now explicitly noted in the limitations paragraph of the Discussion.

Comment 3: No experimental validation of observed transcriptional changes (e.g., functional assays or protein expression validation beyond AbSeq) was conducted.

Answer: Thank you for this comment. While we did not independently validate all antibody reagents included in the AbSeq panel (as most of which were standard, commercially available antibodies with pre-validated specificity) we did perform a targeted verification of the custom-conjugated AbSeq reagents specific to TNFR1 and TNFR2 (TNF-102), which were synthesized specifically for our study. Their performance was benchmarked against conventional fluorochrome-conjugated monoclonal antibodies (anti-TNFRSF1A and anti-TNFRSF1B), demonstrating consistent target recognition and, on average, 7–9% higher detection sensitivity. This confirmed the reliability of these custom reagents for profiling TNF receptor surface expression in our system. The corresponding clarification has been added to the revised Methods section.

Comment 4: The study introduces several candidate biomarkers (CD83, LAMP3, IL24) and immunoregulatory shifts, providing leads for future investigation.

Answer: We thank the reviewer for recognizing the biomarker potential. This perspective has been emphasized in the revised Conclusion.

Comment 5: However, the limited cohort size tempers the impact. Larger, longitudinal studies with functional follow-up are needed to establish clinical utility.

Answer: We agree and have included this in the concluding remarks.

Round 2

Reviewer 1 Report

Comments and Suggestions for Authors

The author has addressed my concern and revised the MS according to my comment. I recommend its acceptance.

Reviewer 3 Report

Comments and Suggestions for Authors

Thank you very much for improving the manuscript. Now it is much better and I think it's ready to publish.